**www.cambridge.org/qrd**

## Perspective

biophysical methods; ion mobility mass spectrometry; machine learning; protein structure analysis; structure prediction

**Corresponding author:**
Michael Landreh;
Email: michael.landreh@icm.uu.se

# Mass spectrometry integrates protein design into structural biology method development

Alexander Stevens[1], Hannah Osterholz[1], Vsevolod Viliuga[2,3], Thibault Vosselman[1], Surabhi Kokane[1], Erik G. Marklund[4], Arne Elofsson[2], Axel Leppert[1] and Michael Landreh[1,5] (ORCID)

[1]Department of Cell and Molecular Biology, Uppsala University, Sweden; [2]Science for Life Laboratory and the Department of Biochemistry and Biophysics, Stockholm University, Sweden; [3]Max Planck Institute for Polymer Research, Mainz, Germany; [4]Department of Chemistry for Life Sciences, Uppsala University, Sweden and [5]Department of Microbiology, Tumor and Cell Biology, Karolinska Institutet, Sweden

## Abstract

Recent advances in machine learning (ML) have transformed protein science, enabling engineering and *de novo* design of artificial proteins with novel structures and functions. However, experimental analysis of key design features, such as oligomerization, folding, ligand binding, and dynamic conformational changes, remains critical. Here, we outline how mass spectrometry (MS) complements protein design through its ability to corroborate a wide range of design objectives. Furthermore, engineered proteins have become valuable tools for exploring the use of MS in detecting structural features, charge effects, and weak interactions by serving as testbeds for method development. Integrating ML and native MS thus creates a feedback loop: new designs challenge analytical techniques, while improved methods provide richer data to guide and improve future predictions. This synergy is vital for expanding the capabilities of protein engineering, including toward applications in synthetic biology and artificial protocell development.

## Introduction

Recent advances in machine learning (ML) have enabled breakthroughs in protein science. A range of ML-based computational tools has been developed as improved solutions to the protein structure prediction problem. These tools, including AlphaFold2 and 3, RosettaFold, ColabFold, and others, utilize deep learning to produce highly accurate structural models (Jumper *et al.*, 2021; Mirdita *et al.*, 2022; Baek *et al.*, 2023; Wohlwend *et al.*, 2025). Besides being able to predict structure from sequence, these advances also revolutionized protein design by being able to predict protein sequences that adopt a specific fold (Goverde *et al.*, 2023; Frank *et al.*, 2024; Pacesa *et al.*, 2024). ML-based tools have enabled co-design of protein structures and interactions with other biomolecules and ligands (Abramson *et al.*, 2024; Baek *et al.*, 2024; Krishna *et al.*, 2024; Butcher *et al.*, 2025). Beyond its impact on the analysis of native proteins and their complexes, these advances have enabled the development of artificial, that is, not naturally evolved, proteins in two ways:

(1)   ML has expanded the horizons of protein engineering. The sequence of an existing scaffold can be modified to accommodate new structural or functional features (Nikolaev *et al.*, 2024; Sumida *et al.*, 2024), and their impact can be assessed using structure predictions. Alternatively, motif scaffolding poses another interesting application where a natural functional structure motif is fixed, and a *de novo* backbone accommodating it is generated.

(2)   *De novo* design of proteins has expanded the protein fold universe, building proteins with new structures using physical, and not evolutionary, constraints. Here, a design objective is set, and new structures are generated using a computational pipeline, which usually has one or more foundational protein ML models at its core.

Both approaches hold great promise for protein science, ranging from the development of new biomaterials to environmental engineering to the targeted modulation of cellular pathways (Chu *et al.*, 2024; Kortemme, 2024; Wang *et al.*, 2024). However, the computational breakthroughs are intricately linked to insights from biophysical protein characterization. The underlying neural networks must be trained on large numbers of experimentally determined structures from the Protein Data Bank, using data from X-ray crystallography, nuclear magnetic resonance, and cryo-electron microscopy. Conversely, experimental methods are also required to corroborate the computationally designed proteins, usually by solving their high-resolution structures with NMR, X-ray crystallography, or cryo-EM.

The implication of the relationship between ML-based predictions and experiment is that the scope of what can be predicted computationally is limited by what can be observed experimentally in sufficiently large quantities to generate training data (Chakravarty *et al.*, 2025). This fact

has largely limited the prediction and design pipelines to stably folded proteins, although advances are being made toward including disorder and dynamics. Strategies such as introducing ambiguity in the structure predictions by clustering or sub-sampling the sequence alignments can help to uncover alternative states in folded proteins (Monteiro da Silva *et al.*, 2024; Wayment-Steele *et al.*, 2024; Kalakoti and Wallner, 2025). For intrinsically disordered regions (IDRs), large-scale molecular dynamics simulations can be used to generate network training data (Tesei *et al.*, 2024) and enable the design of IDRs with specific conformational preferences (Pesce *et al.*, 2024). Disordered and dynamic structures have been included in the design pipelines, such as the *de novo* design of folded binders that induce disorder-to-order transitions in their targets (Liu *et al.*, 2025; Wu *et al.*, 2025), and *de novo* design of enzymes using state-of-the-art all-atom methods (Butcher *et al.*, 2025). Together, these developments have fostered the prediction and design of protein structural dynamics. One example is the design of conformational switches that change their dynamics upon external stimuli such as ion binding (Praetorius *et al.*, 2023; Guo *et al.*, 2025), another is the design of disordered low-complexity sequences that facilitate controlled liquid–liquid phase separation (LLPS) (von Bülow *et al.*, 2025). Combining folded domains that engage in specific interactions with IDRs that mediate dynamic association is a potential step toward cell-like self-organization abilities (Mohanty *et al.*, 2022; von Bülow *et al.*, 2025).

Here, we describe how mass spectrometry (MS) can be used to corroborate protein designs, but also how the concept can be reversed by using designed proteins to test the limits of structural proteomics. Based on these insights, we suggest specifically designed or engineered proteins as a versatile tool to test and expand the limits of biophysical methods.

## Corroborating protein design with MS

Considering the rapid pace of the field, corroborating the increasingly complex designs may become a potential bottleneck for future development. Since the three-dimensional fold predictions are already of high accuracy, experimental investigations are likely to focus on interactions and dynamics. We have previously demonstrated how MS can be integrated with ML-based structure predictions to monitor binding, folding, and stability of proteins (Allison *et al.*, 2022). Although MS generally does not provide sufficient constraints to solve a protein structure from scratch, it can corroborate predicted interactions. Electrospray ionization (ESI)-MS enables the transfer of proteins into the gas phase with minimal distortion of their structural features (Konermann *et al.*, 2013). This approach, termed 'native MS', can be used to measure the masses of intact complexes (Benesch *et al.*, 2007; Lössl *et al.*, 2016). Protein folded states, as well as, in some cases, conformational changes, can be detected by monitoring changes in ion charge state distributions (Konermann *et al.*, 2013; Chingin and Barylyuk, 2018). Native ion mobility MS (IM-MS) can be used to determine the collision cross sections (CCS) of protein ions, which provides insights into their architecture and dynamics (Christofi and Barran, 2023).

MS in combination with protein digestion and liquid chromatography can additionally detect local dynamics through crosslinking or hydrogen/deuterium exchange, which improves structure prediction accuracy (Stahl *et al.*, 2024; Wang *et al.*, 2025). In this manner, MS can capture complex structural processes such as enzyme dynamics, bridging the gap between structure and function (Busenlehner and Armstrong, 2005; Marklund and Benesch, 2019).

However, connecting structural features in the gas phase to predicted or experimental solution structures is not without its challenges. Desolvated protein complexes can self-solvate, where previously solvent-exposed residues form new bonds with neighboring residues or the backbone (Breuker and McLafferty, 2008). Unsupported domains, disordered loops, and cavities can undergo significant compaction when the solvent is removed (Hansen *et al.*, 2018; Rolland *et al.*, 2022). As the entire energy landscape is altered in a vacuum, proteins can adopt entirely new conformations that are not accessible in solution (Cropley *et al.*, 2024). It is important that these effects are considered when comparing MS data to protein structure predictions. Despite these caveats, it was recently found that the use of gas phase CCS data from IM-MS in the scoring function consistently improves the accuracy of AF2 models (Turzo *et al.*, 2022).

The capability to inform about interactions, stabilities, and conformations of proteins makes MS a complementary strategy to ML-based structure prediction. The same applies to *de novo* protein designs, but with one key difference: corroboration of predicted structures largely means confirming oligomeric states and architectures. Validation of a protein design, on the other hand, must confirm the central design objective(s), which can be far more diverse and range from the structural to the functional (Figure 1a, b). MS can detect many designed features: Oligomeric states and ligand binding can be studied with intact mass measurements, interface stabilities can be determined using gas-phase dissociation, the impact of covalent modifications and sequence variations can be validated using native top-down sequencing, and folded-unfolded transitions manifest themselves in changes in charge state distributions or cross sections in ion mobility experiments (Figure 1c). Intact mass measurements have been applied to show the desired stoichiometry of designed protein nanocages (Khmelinskaia *et al.*, 2025). Relative stabilities of hetero-trimers were determined by comparing signal intensities for each species in the mass spectra. Including a surface-induced dissociation (SID) step in the native MS workflow revealed connectivities within mixed oligomers (Chen *et al.*, 2019). By allowing a large number of designed subunits to associate freely in a single mixture and monitoring oligomerization with MS, it was possible to determine which subunits had complementary interfaces without having to solve their structures (Chen *et al.*, 2019; VanAernum *et al.*, 2020). Lastly, the action of a kinase-activated protein switch could be verified by correlating the mass shift from phosphorylation with protein fluorescence (Woodall *et al.*, 2021).

Common to the above examples is that MS is used to check the intended design features. Another advantage is its ability to interrogate multiple features simultaneously (Figure 1d). In this manner, one feature could be changed in the protein, and its impact on another feature can be determined from the resulting mass spectra. A recent example is the use of MS to assess the impact of single-residue substitutions – which is not always reliably predicted by ML (Allison *et al.*, 2022; Buel and Walters, 2022)- on protein oligomerization efficiency (Figure 1d, panel i). By introducing mutations into a designed scaffold and measuring their impact on protein unfolding or dissociation, it is possible to design pairwise interactions that improve stability (Figure 1d, panel i) (Liu *et al.*, 2025). We employed native MS to guide the design of a lipid-binding site in an artificial membrane protein. Here, lipid ligands were added to different scaffold designs, binding was monitored with exact mass measurements, and the impact of the bound lipids on complex stability was determined through gas-phase dissociation (Figure 1d, panel ii). From these experiments, we could delineate that local protein flexibility and the coordination of the lipid give rise to lipid-mediated stabilization of membrane proteins(Abramsson *et al.*, 2025). It is likely that even more

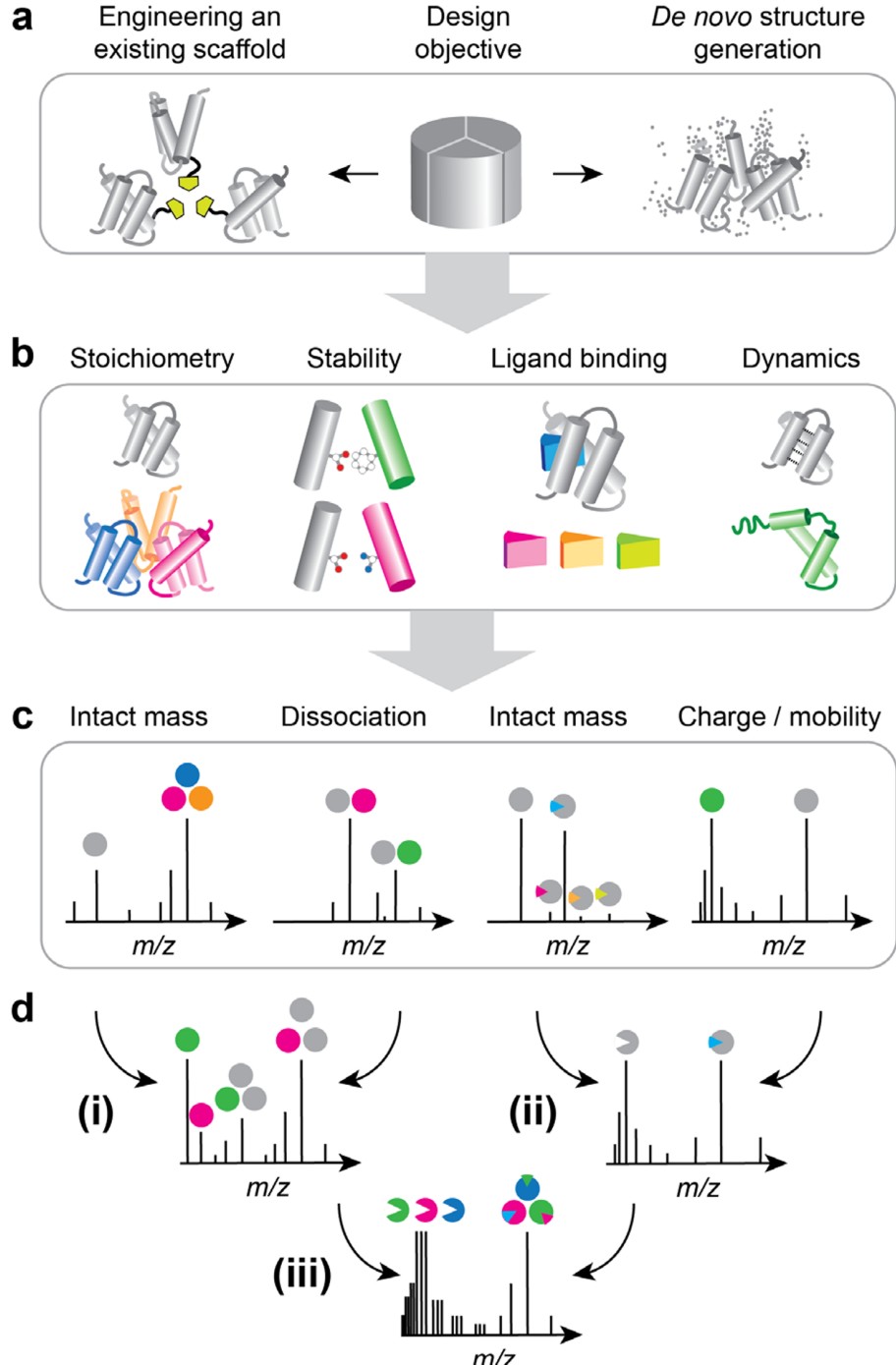

**Figure 1.** Applications of native MS to verify protein design objectives. (a) A design objective (e.g. trimer) is implemented through modification of an existing protein scaffold and confirmed with ML-based structure prediction, or through de novo generation of a structure using generative models. (b) Examples of design objectives that can be probed by MS. (c) Schematic mass spectra showing the manifestation of each feature in (b). (d) Multiple design objectives can be observed in individual mass spectra, such as the impact of mutations on the stability of oligomers (i), ligand-induced changes in protein flexibility (ii), and ligand-induced changes in oligomerization efficiency (iii).

complex combinations of design objectives can be observed by MS, connecting, for example, ligand-controlled changes in flexibility with self-assembly and stability (Figure 1d, panel iii).

## Testing the limits of native MS with engineered proteins

The application of MS to corroborate designs of folded proteins is relatively straightforward, mostly because the abilities and limits of MS have been validated extensively for such systems. Yet some proteins have unusual or complex architectures that test the limits of the approach. Prominent examples are proteins that undergo LLPS, that is, self-assembly into dynamic structures such as stress granules. The underlying interactions are often heterogeneous and can involve an uneven distribution of charged residues, long disordered segments, and multivalent contacts (Wang *et al.*, 2018). MS analysis of these proteins raises multiple questions (Sahin *et al.*, 2023): Can we capture

the presence of multiple folded and/or disordered domains in the same protein? Do extreme solution charges and charge distributions affect ionization? And can MS detect low-affinity interactions between IDRs? Several studies using protein engineering have shed light on how proteins with complex charges, architectures, and interactions can be studied with native MS (Figure 2a).

*Multi-domain proteins.* To find out whether MS can detect the presence of multiple folded domains, Ruotolo and colleagues engineered covalently linked chains of single-domain proteins (Zhong *et al.*, 2014). The resulting chains, containing one to four autonomously folded units, were subjected to solution and gas-phase unfolding and analyzed with IM-MS. Strikingly, the number of unfolding transitions correlated with the number of folded domains, regardless of the proteins that were used as building blocks. These data show that separately folded domains in single protein chains are retained in the gas phase, which suggests that complex protein architectures are amenable to MS analysis (Figure 2b).

*Extreme solution charges.* Native MS relies on ionization of proteins in their native state. For folded proteins, the resulting number of charges scales with their surface area, and charges are carried by ionizable side-chains (Kaltashov and Mohimen, 2005; Kebarle and Verkerk, 2009). Charge states thus provide structural insights (Hall and Robinson, 2012; Chingin and Barylyuk, 2018). However, folded domains in phase-separating proteins can contain extreme or uneven distributions of charged residues (Kapelner and Obermeyer, 2019; Kim *et al.*, 2024). To test whether the number and location of solution charge affects gas-phase structure, we engineered protein variants with positive and negative charges, and scrambled or partitioned their distributions on the surface. We found that surface area dictates electrospray charge in all cases, but partitioning the charge locations destabilizes the protein ions (Figure 2c). These results indicate that the structures and interactions of proteins with uneven or extreme surface charges can be investigated with native MS, provided that Coulombic effects are taken into account (Abramsson *et al.*, 2021).

*Charge and disorder.* Interactions involving highly charged IDRs are a common feature of LLPS (Liao *et al.*, 2024), raising the question of how they are impacted by the electrospray charging process. Barran and co-workers explored the role of charge patterning on IDR conformations in solution and in the gas phase by varying the locations of ionizable residues in an intrinsically disordered protein. Computational simulations and IM-MS showed that proteins with even charge distributions populate dynamic ensembles, whereas partitioning into positive and negative blocks results in increased intramolecular interactions and compaction (Figure 2d).

*Mixed-domain architectures.* Proteins that undergo LLPS often combine folded and disordered domains. We therefore expanded the charge engineering approach to multi-domain architectures, and engineered folded proteins with disordered tails with positive, negative, or no charges. ML was used to generate IDR sequences with comparable solution conformations (Janson *et al.*, 2023). In MS, all variants exhibited near-identical charge state distributions, which correlated with the presumed surface areas of the folded domain with compact, intermediate, and extended IDRs (Figure 2e). Although the correlations were not quantitative, the data suggest that native MS can inform about conformational preferences of IDRs in multi-domain proteins regardless of their charge in solution.

## Artificial proteins as tools for method development

The rapid evolution of protein design opens the possibility of moving toward proteins with unusual structures and properties. However, such designs, which can combine non-specific and specific interactions, disordered and folded domains, and extreme sequence compositions and lengths, are likely to test the limits of experimental corroboration.

Based on the insights that engineered proteins have provided for native MS applications, we propose protein design as a universal tool for the development of biophysical methods. In short, proteins with specific features are designed to test the limits of a specific method, and comparisons between the predicted and experimental data serve as a gauge for method validity. By gradually changing one protein feature at a time and monitoring the impact on the experimental data, it is possible to assess, for example, the sensitivity of a method toward the feature, calibrate its effect on the experimental data, or detect experimental artefacts, paving the way for efficient fine-tuning strategies. The same iterative process can also be applied to test the limits of prediction confidence by defining the point at which prediction and experiment diverge. Some potential applications are described below:

*Defining resolution limits for structural changes and local dynamics.* A recent addition to the protein design toolbox is the generation of proteins with defined local and directional flexibility (Viliuga *et al.*, 2025). This development opens avenues for testing the sensitivity of a method toward local conformational changes or stability. For example, IM-MS analysis of a fixed protein scaffold with varying structural flexibility could be used to determine whether local flexibility affects the stability or CCS of proteins in the gas phase (Figure 3a). An outstanding challenge that could be addressed in this way is the interpretation of gas-phase unfolding data, and whether it can be correlated with structural features in solution (Eldrid *et al.*, 2022; Rider *et al.*, 2024). Going forward, designed protein dynamics could be employed to map the impact of specific thermodynamic properties on binding and folding in the gas phase, for example, using variable temperature (vT) ESI (Laganowsky *et al.*, 2022).

*Model systems for gas-phase structure determination.* Single-molecule structure analysis remains a frontier in biophysics. A promising approach is the use of free electron lasers to create diffraction patterns from individual protein complexes in the gas phase (Neutze *et al.*, 2000), for which (IM)MS is excellently suited to deliver intact or activated mass- and conformation-selected proteins into the X-ray interaction region (Kierspel *et al.*, 2023). This is a powerful concept in its own right, but it requires information about their orientations at the time of exposure. Relative orientations can, in theory, be inferred from the complete data set if the data amount and quality are high enough, but in reality, the algorithms do not always converge to a useful solution. This challenge can potentially be met through electric fields that act on a protein's dipole moment (Marklund *et al.*, 2017; Sinelnikova *et al.*, 2021), although computational simulations highlight the trade-off between field strength, orientation homogeneity, and protein stability (Wollter *et al.*, 2024; Agelii *et al.*, 2025). Additionally, electric fields could be used to control protein unfolding in a precise way (Sinelnikova *et al.*, 2021). Designed proteins with strong dipole moments and stable gas-phase conformations could be used to identify experimental conditions required for structure determination, as they would potentially retain a known structure at high field strengths, yet become sufficiently oriented to generate an interpretable diffraction pattern (Figure 3b).

*'Frozen' intermediates for protein folding studies.* High-resolution protein structures usually represent the native basin of protein folding. Structures of (un-)folding intermediates are short-lived or unstable and therefore considerably more challenging to capture, leading to limited structural information. Protein design potentially

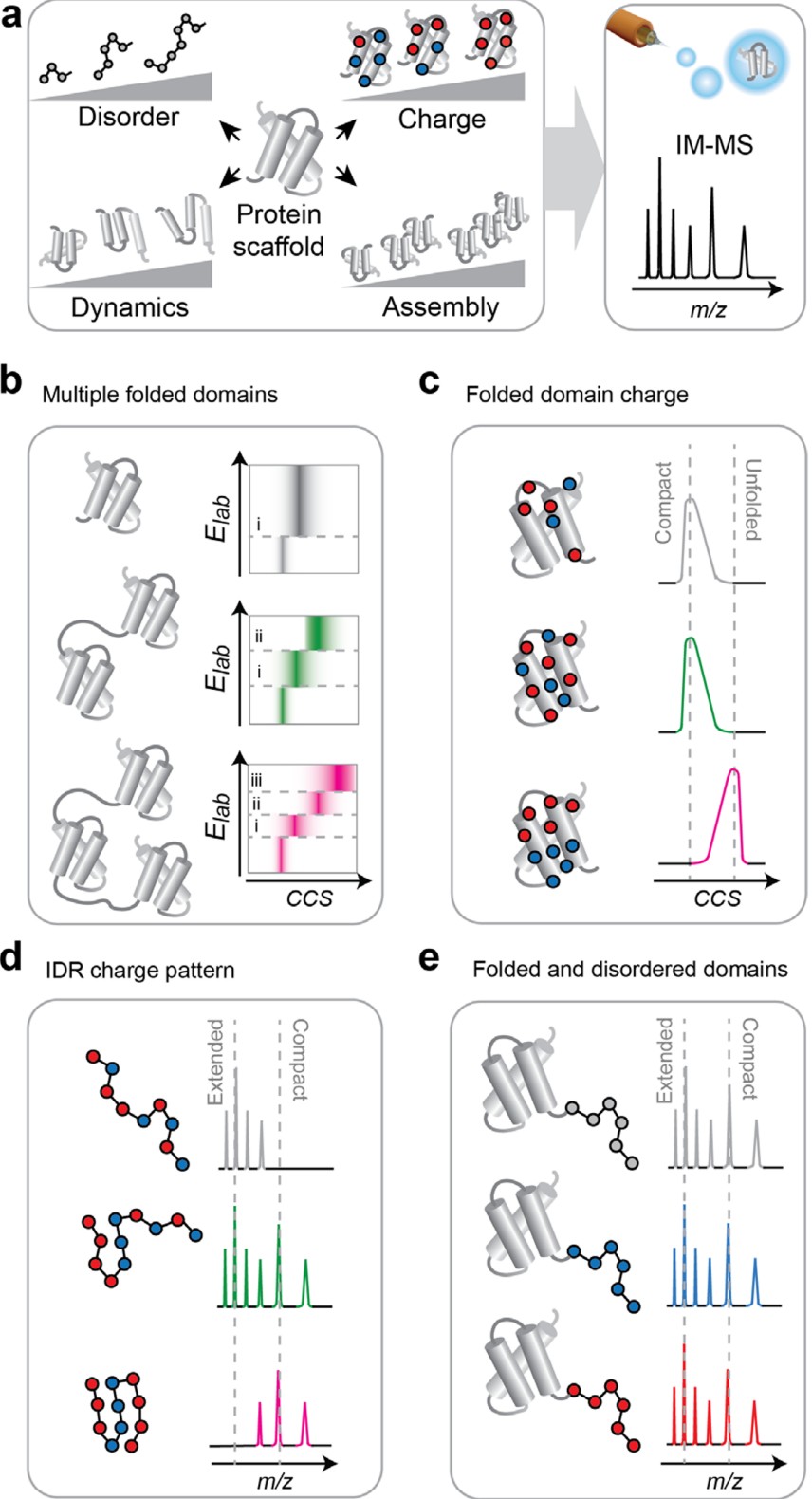

**Figure 2.** Engineered proteins reveal principles of native MS. (a) Protein engineering as a strategy for method testing. Features of interest, such as charged residues, disordered tails, additional domains, or flexible hinges, are added to a protein scaffold. Their effect on electrospray charge and gas-phase conformation is then assessed with native MS and IM-MS. (b) IM-MS captures the presence of multiple folded domains via their unfolding transitions. (c) Conformational stability in IM-MS is impacted by the location of charged residues. (d) IM-MS reveals how the patterning of charged residues in disordered proteins controls compaction in solution. (e) The presence of folded and disordered regions is reflected in ion charge states in MS regardless of protein solution charge.

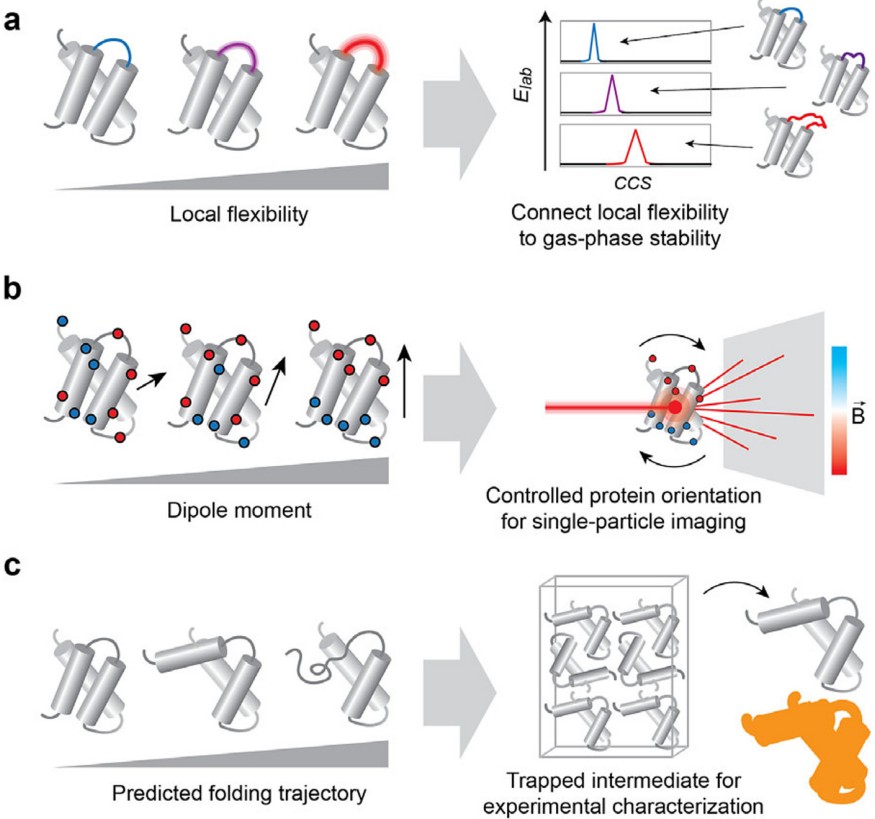

**Figure 3.** Potential applications of protein design for method development in experimental biophysics. (a) Designed proteins with defined local B-factors can be used to test the effect of local structural fluctuations on protein stability and CCS in IMMS experiments. (b) Designed proteins with defined dipole moments can be used to control gas-phase orientation for single-particle imaging with XFEL. (c) Predicted protein folding intermediates can be used as templates for the design of trapped states for high-resolution structure determination. The high-resolution 'trapped state'-structure can be compared to biophysical or computational data on short-lived intermediates (represented as orange shadow).

offers ways to investigate folding pathways. For example, predicted intermediate states could be used as design templates to create 'frozen' snapshots, or scaffold proteins could be generated to stabilize a specific intermediate for biophysical analysis. These reference data could be employed to corroborate experimental observations or to deconvolute overlapping states (Figure 3c).

## Conclusions

The ever-increasing accessibility of protein design tools continues to expand their applications in biomaterial science, biotechnology, and biomedicine. Proteins can be designed for a variety of purposes, but also with previously inaccessible properties relating to their charge, stability, dynamics, size, and shape. As outlined here, their adaptability makes them excellent tools for testing the limits of MS. The fact that the pipeline from design to recombinant material is relatively short and uncomplicated means that proteins can be designed and produced locally in time- and resource-effective ways. They also offer a way to standardize or compare methods across instrument platforms or labs by potentially reducing issues with variability and availability that can be encountered with naturally occurring proteins. Therefore, we believe that proteins that have been engineered or designed with ML can help expand the biophysical toolbox by providing bespoke systems for method development. We anticipate that artificial proteins will emerge as standard reagents for a variety of structural biology applications.

**Open peer review.** To view the open peer review materials for this article, please visit http://doi.org/10.1017/qrd.2025.10018.

**Acknowledgments.** The authors thank their colleagues for the helpful discussions.

**Author contribution.** Conception: AS, AE, EGM, AL, ML. Original draft writing: AS, HO, VV, ML. All authors contributed to the final article.

**Financial support.** ML is supported by a KI faculty-funded Career Position, a Cancerfonden Project grant (22–2023 Pj), a Swedish Research Council (VR) Project Grant (2024–04483), and a Consolidator Grant from the Swedish Society for Medical Research (SSMF). AE is supported by a KAW Project grant (2022.0032). EGM is supported by a Swedish Research Council (VR) Project Grant (2020–04825) and a Röntgen-Ångström Cluster Grant (2021–05988).

**Competing interests.** The authors have no conflicts of interest to declare.

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
