## [Reviewer Report]

The manuscript provided a nice summary connecting protein design with native mass spectrometry. It is overall well written, I have a few suggestions to add several relevant points into the discussion:

1.Figure 1: the disscussion on dissociation has only been focused on breaking of noncovalent interations. Native top-down has demonstrated potential to provide residual level structural information, especially with UVPD and ECD.

2.Native MS studies on dynamic changes of proteins, enzyme reactions, should be mentioned because they help connect structures with function.

3.Native MS can also provide important biophysical parameters, especially with variable temperature nESI source. These studies may be mentioned with Figure 2 to show how native MS can provide quantitative structural data.

---

## [Reviewer Report]

This is an interesting and timely opinion piece, covering recent advances in ML-based protein design and its relationship to native MS experiments. Listed below are a few points that the authors should address when preparing the final version of their manuscript.

Specific Comments and Suggestions:

Abstract: consider using the word corroboration/corroborate only once.

Abstract and Introduction: should clarify that electrospray is the method of choice for native ESI (this may not be obvious for non-experts)

Introduction, mention specific examples of ML tools (such as Alphafold), instead of just talking about ML in vague terms.

p. 2: do not omit NMR from 2nd last sentence.

p. 3 top: the text switches from ML to AI. It’s probably best to only use ML. OR specify your reasons for using one vs the other, and clearly define the difference.

Consistently use the MS acronym once it has been defined, do not go back and forth between MS and “mass spectrometry” (same with ML vs “machine learning”).

p. 4 “Large-scale conformational changes can be detected via ion mobility mass spectrometry (IM-MS) (Christofi & Barran 2023)” Here and elsewhere, the authors seem to imply that it can always be assumed that solution structure = gas phase structure. This may or may not be true, and the ongoing discussion should be highlighted. Relevant papers include:

Breuker, K.; McLafferty, F. W. Stepwise evolution of protein native structure with electrospray into the gas phase, 10-12 to 102 s. Proc. Natl. Acad. Sci. U.S.A. 2008, 105, 18145-18152.

Hansen, K.; Lau, A. M.; Giles, K.; McDonnell, J. M.; Struwe, W. B.; Sutton, B. J.; Politis, A. A Mass-Spectrometry-Based Modelling Workflow for Accurate Prediction of IgG Antibody Conformations in the Gas Phase. Angew. Chem.-Int. Edit. 2018, 57, 17194-17199.

Williams, R. V.; Huang, C.; Moremen, K. W.; Amster, I. J.; Prestegard, J. H. NMR analysis suggests the terminal domains of Robo1 remain extended but are rigidified in the presence of heparan sulfate. Sci. Rep. 2022, 12, 14769.

Cropley, T. C.; Liu, F. C.; Chai, M. Q.; Bush, M. F.; Bleiholder, C. Metastability of Protein Solution Structures in the Absence of a Solvent: Rugged Energy Landscape and Glass-like Behavior. J. Am. Chem. Soc. 2024, 146, 11115-11125.

Some of Lindert’s work should be mentioned and cited, such as

Turzo, S. M. B. A.; Seffernick, J. T.; Rolland, A. D.; Donor, M. T.; Heinze, S.; Prell, J. S.; Wysocki, V. H.; Lindert, S. Protein shape sampled by ion mobility mass spectrometry consistently improves protein structure prediction. Nat. Commun. 2022, 13, 15.

p. 7: “phase-separating proteins” unclear. Please explain.

p. 7: “uneven surface charges” unclear. Please explain.